# Comparing the effects of interactive and conventional video education on activation, treatment adherence, and weight changes in dialysis patients: A randomized clinical trial protocol

Sogand Sarmadi[1], Neda Sanaie[2], Akbar Zare-Kaseb [1]*

**1** Department of Medical-Surgical Nursing, Shahid Beheshti University of Medical Sciences, Tehran, Iran,
**2** Department of Medical Surgical Nursing, School of Nursing and Midwifery, Shahid Beheshti University of Medical Sciences, Tehran, Iran

* akbar.zarekaseb@gmail.com

## Abstract

### Background

Patients with end-stage renal disease undergoing hemodialysis face substantial challenges in adhering to complex therapeutic regimens, significantly impacting morbidity, mortality, and quality of life. While conventional educational methods offer some benefit, interactive digital tools may yield deeper engagement and sustained behavioral change.

### Objective

This study assesses and compares the impact of interactive and conventional video-based education, as well as usual care, on patient activation, treatment adherence, and inter-dialytic weight gain in individuals undergoing hemodialysis.

### Methods

A three-arm, parallel-group, randomized clinical trial will be conducted in three academic hospitals in Tehran, Iran. A sample of patients will be enrolled and distributed into one of three categories: (1) interactive video education, (2) conventional video education, or (3) typical nurse-led education. The 13-item Patient Activation Measure will be used to measure the primary outcome of patient activation. Secondary outcomes will encompass treatment adherence, as measured by the End Stage Renal Disease Adherence Questionnaire, and inter-dialytic weight gain. Evaluations will occur at baseline, immediately post-intervention, and at 1- and 3-month follow-up intervals. Data will be analyzed using intention-to-treat principles with mixed-effects modeling.

**Data availability statement:** No datasets were generated or analysed during the current study.

**Funding:** The author(s) received no specific funding for this work.

**Competing interests:** NO authors have competing interests.

## Discussion

This trial is among the first to rigorously compare interactive and conventional video education in a dialysis population. Findings may inform scalable, cost-effective strategies for improving self-management and adherence in patients with end-stage renal disease. This protocol was registered prospectively at ClinicalTrials.gov (Registration No. NCT07099326) on July 31, 2025. The National Research Ethics Committee also approved the study with the ethics code: IR.SBMU.PHARMACY.REC.1404.067.

## Introduction

Chronic kidney disease (CKD) is a significant global health risk factor and is more prevalent in people over 60 years old [1]. CKD denotes a state of impaired renal function characterized by the loss of over half of normal kidney function [2], and is identified by indications of kidney damage or an estimated glomerular filtration rate (eGFR) below 60 mL/min/1.73 [3]. Based on eGFR values, CKD patients are stratified into five distinct stages [4]. Advanced stages (3–5) of the disease significantly affect daily activities, nutritional status, fluid and electrolyte balance, and overall health, potentially leading to uremic syndrome and mortality if untreated [5]. End-stage CKD, also termed End-Stage Renal Disease (ESRD), involves an irreversible decline in kidney function, causing continuous dialysis or kidney transplantation for survival [6].

According to 2016 statistics from Iran, there were over 55,000 documented CKD patients, including 27,500 receiving hemodialysis (HD) and 1,600 receiving peritoneal dialysis (PD) [1]. Further research shows that the number of CKD patients in Iran rises by approximately 15% each year [1,7]. At present, HD represents the predominant treatment modality for ESRD within the nation [8]; however, individuals undergoing HD encounter challenges, such as the imperative of strict dietary adherence to mitigate cardiovascular complications [9]. For an HD program to be implemented effectively, four critical aspects must be followed: diet, medication, fluid restriction, and consistent dialysis session attendance [1].

As demonstrated by their behavior [10], the patient's active participation in treatment, adherence to recommendations, and receipt of care services define treatment adherence. Multiple studies indicate that patients undergoing HD show suboptimal adherence to treatment plans, which can accelerate disease progression and increase the risk of emergency hospitalizations [11–13]. As biochemical and physiological indicators, Kt/V and inter-dialytic weight gain (IDWG) are used to assess dialysis adherence [10,14–16].

To improve adherence and self-management among patients afflicted with chronic illnesses, the concept of "patient activation" was developed, measuring the patient's knowledge, skills, and confidence in managing their own health via the PAM (Patient Activation Measure) tool [17]. Increased activation shows a relationship with optimal treatment consequences, a reduction in unwarranted emergency care utilizations, and reduced rates of hospital re-attendance [18]. Patient activation can be modified and is influenced by elements including age, gender, socioeconomic standing, CKD

stage, disease duration, and existing comorbidities [17,19]. This concept is classified into four levels: level 1, at which the patient experiences being overwhelmed by the disease and a lack of motivation; Level 2, where the patient has begun self-care but lacks adequate knowledge and confidence; At the third level, the patient seeks to maintain and improve health, yet lacks the abilities and beliefs, and at level 4, the patient shows sufficient knowledge and skill for actively managing their disease [20]. Research shows a direct correlation between the degree of patient engagement and adherence to dietary and fluid limitations; diminished engagement may cause non-adherence and subsequent weight increase between dialysis treatments [21,22].

The difference between a patient's weight at the close of a dialysis session and the subsequent session's beginning, known as IDWG, is deemed a meaningful measure of fluid regulation and, secondarily, patient adherence to fluid restrictions and clinical factors [14]. Despite international guidelines recommending that IDWG be maintained below 4–4.5% of dry weight, many patients cannot adhere to this restriction and consequently experience excessive weight gain between dialysis sessions [16,23]. Many investigations have shown that heightened IDWG correlates with an elevated incidence of mortality and cardiovascular morbidity, encompassing left ventricular hypertension and unfavorable cardiovascular and cerebrovascular occurrences [24–26]. Excessive weight gain also results in a higher frequency of dialysis treatments, which considerably impairs one's quality of life (QoL) and substantially elevates healthcare expenses [27,28]. Significant impediments to compliance with dietary and fluid limitations encompass an insufficient understanding of fluid balance, diminished incentive, and pre-existing pathological states affecting residual renal performance and fluid elimination rates [27,29].

Educational training forms an essential component of nursing practice, executed via various modalities [30]. Conventional video-based education is one such method, where patients view pre-recorded videos and receive information on the topic in a one-way manner [31]. Research shows that these videos may enhance patient comprehension of disease-related care and ease pre-treatment anxiety [31,32]. An alternative approach is education delivered via interactive video. Patient engagement with the material through interactive questions, virtual simulations, and immediate feedback promotes deeper learning and improves motivation to comply with treatment recommendations, in contrast to passive video observation [33,34].

Conventional video education reliably increases patient knowledge in chronic illness settings, including preliminary studies in dialysis, but systematic reviews report that knowledge gains do not consistently translate into improved clinical or utilization outcomes; the specific educational features that drive behavior change are not well defined. Importantly, there is a paucity of rigorous randomized clinical trials (RCTs) in the hemodialysis population that compare interactive video formats (with real-time feedback, embedded assessments, or simulated practice) against passive, conventional videos for their effects on patient activation and objective adherence measures [35]. Interactive, adaptive educational modalities have improved engagement and some behavioral or clinical outcomes in other chronic-disease contexts, and patient activation (measured by PAM) is a promising mediator of sustained self-management; however, the relationship between activation-focused, interactive education and dialysis outcomes (e.g., Kt/V, session attendance, and IDWG) remains inadequately tested [36]. Because elevated IDWG is consistently associated with greater cardiovascular risk and mortality in dialysis cohorts, an RCT comparing interactive vs conventional video education is timely and necessary to inform evidence-based educational strategies that aim to change behavior and improve clinically meaningful endpoints. Therefore, designing an RCT comparing the effectiveness of interactive video education with that of passive video regarding patients' activation variables, treatment adherence, and weight changes may lead to more effective solutions and improvements in the quality of care for these patients.

## Method

This protocol has been developed in accordance with the SPIRIT 2025 (Standard Protocol Items: Recommendations for Interventional Trials) guidelines to ensure transparency, completeness, and reproducibility of the trial design (S1 Table) [37].

## Trial design

This study is designed as a three-arm, parallel-group RCT with a 1:1:1 allocation ratio. Hospitals (Shohada-ye Tajrish, Taleghani, and Imam Hossein) are the desired study settings. The trial will be conducted within a superiority framework to compare the effectiveness of the educational interventions across the study arms.

## Setting

Participants will be recruited from inpatient units in three hospitals—Shohada-ye Tajrish, Taleghani, and Imam Hossein—all affiliated with Shahid Beheshti University of Medical Sciences in Tehran, Iran. Recruitment will occur within hospital clinics and wards to ensure appropriate access to the target population.

## Sample size

Due to the absence of previous studies closely matching the design and interventions of the present trial, the sample size calculation will be based on data obtained from a preliminary pilot study that the research team will conduct. A pilot study will be conducted in the same dialysis units as the definitive three-arm trial. It aims to (1) estimate parameters required for sample-size calculation (primary-outcome SD), (2) assess recruitment and retention rates, (3) measure patient adherence and provider fidelity, (4) evaluate methodological contamination, and (5) identify operational or data-quality problems.

The selection of this approach ensures that the sample size possesses sufficient statistical power to identify clinically relevant differences between groups, taking into account the unique features of both the study population and the interventions.

Participants will be screened consecutively using the same inclusion/exclusion criteria as in the primary protocol, provide informed consent, and be randomized to one of three arms. The intervention principles, outcome instruments, and timepoints will be identical to those in the definitive trial. For feasibility-only objectives, approximately 12 participants per arm (36 total) will be recruited [38]; if variance estimation is required, around 35 per arm (105 total) will be targeted [38].

A pilot report will document parameter estimates and operational findings. Pilot outcomes include: Standard Deviation (SD) and 95% Confidence Interval (CI) of the primary outcome, recruitment and retention proportions, adherence categories, fidelity scores from structured checklists, contamination based on self-report and digital access logs, and data-quality indicators (missing data, questionnaire burden, technical issues).

Pilot estimates will inform the definitive sample size, attrition-adjusted recruitment targets, and possible design modifications if contamination is substantial. Any changes to the main trial will be justified, approved, and reported. The sample size will be estimated using the pilot data, with a significance level of 0.05 (two-sided) and a statistical power of 80%. The total estimates will be allocated equally across the three study groups. By relying on pilot data without prior literature, the study design ensures methodological rigor and optimizes resource use while maintaining sufficient power to address the primary research objectives.

## Eligibility criteria

Eligible participants will be required to meet all of the following criteria: (1) willingness to participate in the study, (2) literacy in reading and writing, (3) alertness and orientation to time, place, and person sufficient to respond to questions, (4) no history of auditory or visual impairments, (5) absence of cognitive disorders, (6) ownership of a personal mobile device or equivalent capable of running interactive video applications and ability to use it, (7) no current use of psychoactive medications, (8) confirmed diagnosis of CKD by a nephrology specialist with an active medical record in the dialysis unit, and (9) age between 18 and 65 years. Participants will be withdrawn from the study upon meeting any of the following exclusion criteria: (1) voluntary withdrawal from the study at any time, (2) failure to receive and engage with the assigned

video interventions, (3) death of the participant, and (4) transfer to a healthcare facility affiliated with universities other than Shahid Beheshti University of Medical Sciences.

## Ethics approval and consent to participate

The researcher obtained the code of ethics to carry out this research (IR.SBMU.PHARMACY.REC.1404.067) from the Ethics Committee of Shahid Beheshti University of Medical Sciences, registered in ClinicalTrials.gov under the code NCT07099326, and received permission to conduct the research from the Research Vice-Chancellor of Shahid Beheshti University of Medical Sciences All the participants will be provided written informed consent and will be notified of the right to withdraw from participation at any time during the research until publication. Data confidentiality will be ensured, and the results will be provided to the participants at their request. Also, we confirm that all experiments will be performed per the relevant guidelines and regulations. The ethical principles of the Declaration of Helsinki will be followed throughout the study.

## Status and timeline of the study

Recruitment for the participants is planned to begin in September 2025. After recruiting, the intervention and its follow-up are expected to be completed by February 2026. The data analysis will be conducted in February 2026, and the manuscript will be completed by March 2026. The SPIRIT schedule of enrollment, interventions, and assessments is presented in Fig 1.

## Interventions

**Educational content.** A comprehensive literature review will be conducted to identify the optimal educational content. This process will involve the extraction of relevant articles, books, programs, and clinical and academic guidelines. A systematic search will be conducted of English-language databases, namely PubMed, Scopus, Web of Science, and Embase. SID, Magiran, and Iranmedex will also be explored among Persian-language databases. The searches will be performed using keywords including "Patient Activation," "Treatment Adherence," "Hemodialysis," as well as their synonyms and alternative terms. The search time-frame will span from the inception of each database until June 2025.

Subsequently, the initial program content will be developed and then evaluated. Five dialysis-experienced nurses will assess the program's comprehensiveness through a checklist and pinpoint areas needing further development; their feedback will be incorporated into the final version of the program. After the educational program is finalized, its content will be validated by ten nursing faculty members and nephrologists. This panel will be purposively selected to validate the program content. Eligibility criteria will include an academic appointment in nephrology nursing, dialysis care, or patient education; at least five years of relevant clinical or teaching experience; and prior involvement in curriculum development or content validation. Faculty will be recruited through institutional records and professional networks, and all will provide independent ratings of content relevance and clarity using a standardized checklist. Based on the experts' evaluations, supplementary materials will be produced, and necessary revisions will be made.

A Content Equivalence Checklist will be developed by the research team and independently verified by two subject-matter experts. The checklist will include items such as (a) verifying each predefined knowledge unit is included, (b) following the same clinical case vignettes, (c) using equivalent terminology and explanations, and (d) maintaining consistency in visual aids and key messages. Both video versions will be pilot-tested against this checklist, and inter-rater agreement will be reported. Any discrepancies will be resolved through consensus before final approval.

We will prospectively estimate the typical contact time under interventions and usual care to facilitate transparent comparison of exposure across arms. All actual contact times (minutes per session and number of contacts) will be logged prospectively for every participant and reported in the primary results.

| | TRIAL PERIOD | | | | | | | |
|---|---|---|---|---|---|---|---|---|
| | Enrolment | Allocation | Post-allocation | | | | | Close-out |
| TIMEPOINT | $-t_1$ | 0 | $t_1$ | INT | $t_2$ | $t_3$ | $t_4$ | $t_x$ |
| ENROLMENT: | | | | | | | | |
| Eligibility screen | X | | | | | | | |
| Informed consent | X | | | | | | | |
| Allocation | | X | | | | | | |
| INTERVENTIONS: | | | | | | | | |
| Interactive Video | | | | ▬▬ | | | | |
| Conventional Video | | | | ▬▬ | | | | |
| Control | | | | ▬▬ | | | | |
| ASSESSMENTS: | | | | | | | | |
| PAM | | | X | | X | X | X | |
| ESRD AQ | | | X | | X | X | X | |
| IDWG | | | X | | X | X | X | |
| DATA ANALYSIS: | | | | | | | | X |

**Fig 1. SPIRIT schedule of enrolment, interventions, and assessments.** $-t_1$ = Immediately Before Allocation, $t_1$ = Immediately Before Intervention, $t_2$ = Immediately After Intervention, $t_3$ = 1-Month After Intervention, $t_4$ = 3-Month After Intervention, INT = Intervention Period, 0 = Allocation Period, $t_x$ = After the End of Follow-ups.

## Interactive video education arm

The educational intervention will be delivered in two sequential phases over a total duration of six hours. A cohort of participants will be enrolled and organized into groups of five, each group facilitated by a certified instructor with expertise in interactive video pedagogy.

**Phase 1: Interactive video–based instruction (10 × 30-minute modules).** Participants will log in individually to the secure platform to complete ten interactive modules focused on ESRD pathophysiology and alternative treatment modalities. One 30-minute module will be released per day. Context-sensitive pop-up queries will prompt participant responses during each module, and divergent answer pathways will dynamically determine subsequent scenario content. Embedded multiple-choice assessments will further evaluate comprehension; all responses will be recorded in real time via the learning management system for instructor review. Before module commencement, participants will undergo a standardized orientation session to ensure operational proficiency with the platform and clarity regarding study procedures.

**Phase 2: Consolidation and synthesis workshop (1 hour).** Upon completing the interactive modules, participants will reconvene in their assigned groups for a one-hour, instructor-led workshop. The instructor will facilitate critical discussion of the scenarios, guide the extraction of salient clinical concepts, and synthesize group findings. This workshop will emphasize higher-order cognitive processing of disease complexity and integration of theoretical constructs. The session will conclude with targeted didactic feedback to reinforce key learning objectives and enhance long-term knowledge retention.

## Conventional video education arm

The educational intervention will be delivered in two sequential phases over a total duration of six hours. Participants will be enrolled and organized into cohorts of five participants, each facilitated by an experienced instructor conversant with video-based teaching methods.

**Phase 1: Conventional video-based instruction (10×30-minute modules).** Participants will access the secured learning platform at https://spotplayer.ir/ to view ten pre-recorded video modules detailing ESRD pathophysiology and alternative treatment modalities. One 30-minute module will be released daily. Videos will be streamed in their entirety without interactive elements; the instructor will oversee module completion and address any technical queries but will not interject during playback.

**Phase 2: Consolidation and synthesis workshop (1 hour).** After completing all video modules, participants will reconvene in their assigned cohorts for a one-hour instructor-led workshop. The instructor will facilitate group analysis of the presented content, guide extracting key clinical concepts, and synthesize collective insights. This session will foster critical reflection on disease complexity and integration of theoretical knowledge. The workshop will conclude with targeted didactic feedback to reinforce the principal learning objectives and support retention.

## Control group (Usual care)

Participants in the control arm will receive the standard educational practices routinely implemented at each site. Specifically, this will include:

1. **Nurse-led education:** Registered nurses in the dialysis unit will provide face-to-face instruction on disease management and self-care behaviors.

2. **Educational pamphlets:** Participants will be given institution-approved printed materials covering key aspects of ESRD and alternative therapies.

3. **Nurse follow-up:** A dedicated follow-up nurse will provide regular monitoring, either in person or via telephone, to reinforce educational information, address inquiries, and assess adherence to suggested self-care methods.

All procedures will be conducted according to each hospital's established protocols and within the time-frame of the three-hour educational session allocated to the intervention arms (Fig 2).

## Randomization, allocation concealment, and implementation

Participants will be randomly assigned to one of the three study arms (1 = interactive video, 2 = conventional video, 3 = usual care) in a 1:1:1 ratio.

## Sequence generation

An independent biostatistician, who is not involved in patient enrollment or intervention delivery, shall produce a computer-generated randomization list. Randomization will be stratified by hospital and dialysis shift to ensure balanced assignment across recruitment sites. Within each stratum, variable block sizes—such as 3, 6, and 9—will be employed to reduce predictability while maintaining allocation equilibrium. The sequence shall be generated utilizing the 'blockrand' package in R (version 4.x) with a reproducible seed.

## Allocation concealment

Assignments will be implemented through a centralized, password-protected web-based randomization system to maintain allocation concealment and prevent selection bias. The allocation will only be revealed after participant eligibility and

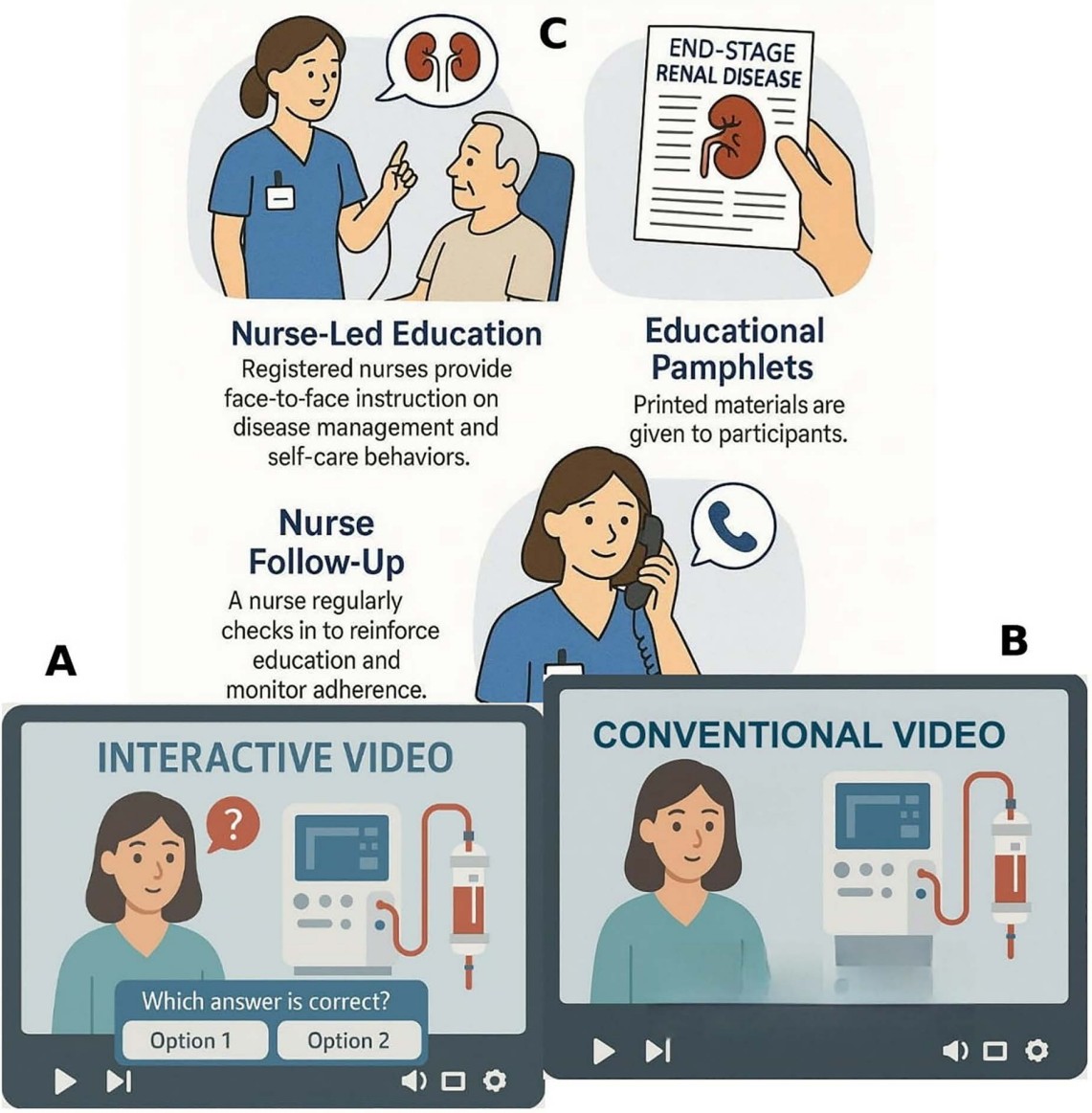

**Fig 2. Interventions schematic details.** A: Interactive Video, B: Conventional Video, C: Usual Care.

baseline assessment have been completed. If a central system is unavailable, sequentially numbered, opaque, sealed envelopes, prepared by the independent statistician, will be used following best practices [39].

## Implementation

The allocation sequence shall be generated and stored securely by the independent statistician. Research nurses responsible for screening and enrolling participants cannot access the sequence. After confirming eligibility and obtaining informed consent, the enrolling investigator will log into the central randomization platform to acquire the group assignment. Intervention facilitators and outcome assessors shall remain blinded to the randomization list, and outcome assessors shall be blinded to group allocation.

## Blinding

Given the nature of the educational interventions (interactive video, conventional video, and standard nurse-led education), neither participants nor instructors can be blinded to group assignment. Outcome assessors will be blinded to intervention allocation when administering follow-up questionnaires and extracting clinical and adherence data from medical records. The trial's statistical programming team will conduct all analyses on a de-identified dataset in which intervention arms are labeled as Group A, B, or C. The key linking these labels to the actual interventions will be held securely by the trial statistician and not revealed until after the primary analyses are complete.

## Permissible Unblinding

Unblinding of data analysts will only occur under exceptional circumstances in which knowledge of the participant's educational assignment is required to address a serious concern. The principal investigator must receive all unblinding requests in writing and will subsequently consult with the Data and Safety Monitoring Board. If approved, the trial statistician will disclose the specific participant's allocation from the master randomization file. The study records will document all unblinding events, including the justification, date, and personnel involved.

## Contamination control and monitoring

Considering the standard setting of dialysis units and the potential for information exchange among patients, there exists an intrinsic risk of contamination between study groups. A combination of procedural and analytical strategies will be implemented to address this concern.

## Secure delivery of video content

Educational videos associated with individual user accounts shall be provided via a secure, password-protected online platform. Access and viewing durations will be systematically recorded for each participant to uphold integrity and identify any potential unauthorized cross-use.

## Separate workshops and facilitators

Consolidation workshops shall be scheduled separately for each study arm, with dedicated facilitators appointed to each group. Facilitators will be instructed to refrain from sharing educational materials across groups.

## Monitoring of cross-exposure

At each subsequent follow-up appointment, participants will be inquired as to whether they have been exposed to educational materials from other study groups (e.g., by watching videos or receiving information from peers). These self-reports will be utilized to estimate the rate of contamination.

## Sensitivity analyses

If the observed contamination rate exceeds 20%, the potential for dilution of intervention effects will be clearly reported. Sensitivity analyses will be performed to evaluate the robustness of findings under different assumptions about contamination.

## Strategies to mitigate instructor bias, workshop imbalance, and participant adherence

Several fidelity procedures will be implemented to minimize potential instructor/facilitator bias and ensure that observed differences are attributable to the interactive video components rather than variations in facilitator performance.

A facilitator manual will be developed, containing the exact script for each workshop, learning objectives for each section, guiding questions, and a structured activity timeline. A fidelity checklist will be completed by the facilitator after each session and verified by the study monitor; checklist items will include adherence to the script, duration of response time, avoidance of supplemental content, and documentation of participant attendance. All facilitators will undergo standardized training and competency testing before study initiation, with a required passing score of ≥85% on the procedural assessment. In addition, selected sessions will be audio- or video-recorded for independent fidelity review by three external evaluators blinded to allocation, and corrective feedback will be provided during the first ten sessions.

Participant engagement with the pre–recorded videos will be supported by automated SMS or telephone reminders sent one hour before each scheduled module release. Video-access logs within the learning management system will capture start and completion times for each 30-minute session and quiz completion rates where applicable. Nursing staff in the control arm will analogously document each face-to-face or telephone education encounter's date, duration, and content.

A per-protocol definition will classify participants as "treated as planned" if they complete at least 100% of their assigned educational components delivered by certified facilitators (i.e., ten of ten video modules). Participants failing to meet this threshold or whose sessions are conducted by non-certified staff will be categorized as "not treated as planned."

## Outcome measures

**Primary outcome.** Patient activation will be measured using the 13-item Patient Activation Measure (PAM) created by Hibbard et al. [40]. Participants will rate each item on a five-point Likert scale— "strongly disagree" (1), "disagree" (2), "agree" (3), "strongly agree" (4), or "not applicable" (no score)—yielding a raw total score between 13 and 52 [41]. Raw scores will be transformed to a standardized 0–100 scale per the PAM scoring algorithm and classified into four activation levels:

- Level 1 (disengaged and overwhelmed): ≤ 47

- Level 2 (becoming aware of the need for self-management): 47.1–55.1

- Level 3 (taking action): 55.2–67

- Level 4 (maintaining behaviors and pushing further): > 67.1

The English-language PAM has demonstrated excellent reliability in nephrology populations (Cronbach's α = 0.87) [40]. The Persian version of PAM demonstrated a Cronbach's alpha coefficient of 91% [20]. For this trial, a Persian translation will be produced via forward- and back-translation by a bilingual expert and undergo content validation by ten nursing faculty members. Internal consistency of the Persian PAM will be assessed within our study cohort by calculating Cronbach's alpha. PAM assessments will be conducted at baseline and at each follow-up visit to evaluate changes in patient activation over time.

**Secondary outcome.** Patient adherence to prescribed therapy will be evaluated using the End-Stage Renal Disease Adherence Questionnaire (ESRD-AQ), a validated 46-item self-report instrument encompassing five domains: General ESRD Knowledge (5 items), Hemodialysis Attendance (14 items), Medication Compliance (9 items), Fluid-Restriction Adherence (10 items), Dietary Recommendation Compliance (8 items). The items utilize a combination of Likert scales, multiple-choice formats, and dichotomous (yes/no) responses. Total scores range from 0 to 1,200, with higher values reflecting greater overall adherence [42]. The ESRD-AQ has demonstrated excellent psychometric properties: Rafiee et al. (2014) reported a Cronbach's α of 0.91 and test–retest reliability of 0.85 [43], while Kim et al. (2010) established a content validity index (CVI) of 0.99 [42]. In this trial, the ESRD-AQ will be administered at baseline and at each follow-up visit to quantify changes in patient adherence over time.

Also, in this study, IDWG will be measured as a secondary outcome by weighing patients immediately before connecting to the dialysis machine and right after each HD session, following a set of standardized protocols. All weightings will be performed using a calibrated medical digital scale (calibrated monthly). Patients will always be weighed wearing a standardized light dialysis gown, emptying their bladder beforehand, and refraining from food or drink before weighing. Additionally, all personal items (shoes, bags, jewelry) will be removed before weighing. Trained staff or the researcher will record all measurements at fixed times (before starting and after finishing dialysis) meticulously to ensure data consistency and allow reliable comparison of weight changes throughout the treatment period [24].

## Data collection and management

The principal investigator will oversee trained research assistants in collecting all trial data, which will be gathered via structured and validated instruments. The primary and secondary outcomes will be assessed via self-reported questionnaires completed by participants at baseline (pre-intervention), immediately post-intervention, and at 1–3 months' follow-up (Fig 3).

Data will be collected in paper-based format at the clinical sites and subsequently entered into a secure electronic database. Double data entry will be employed to minimize transcription errors. All data collectors will undergo a standardized training session before study initiation to ensure uniformity in data collection procedures and administration of outcome measures. A structured questionnaire will be used to collect baseline data, consisting of two main sections:

**Section 1: Demographic data.** This section considers age, sex, educational level, marital status, employment status, place of residence, duration from CKD diagnosis, duration from HD initiation, and presence of comorbid conditions.

**Section 2: Clinical data.** Clinical variables will include daily salt intake (in grams), daily urine output (in milliliters), number of HD sessions per week, amount of fluid intake, and any dialysis-related complications. These data will be extracted from the participants' medical records before the intervention.

## Plans to promote participant retention and complete follow-up

To maximize retention and reduce missing data, the following strategies will be implemented: Research staff will maintain monthly contact with participants via phone or messaging to encourage continued participation and address any technical or logistical challenges. Also, participants will receive automated reminders (SMS or calls) for each video session and follow-up assessment point. Assessments will be conducted during routine dialysis sessions to reduce participant burden. In the event of participant discontinuation or deviation from the intervention protocol, all primary and secondary outcome data will be gathered at the defined follow-up intervals. Reasons for discontinuation or protocol non-adherence will be documented through structured exit interviews. All outcomes will be analyzed under the intention-to-treat principle. Missing data (overall percentage, distribution across arms, and patterns) will be carefully described. If the amount of missing data on the primary outcome is very low ($\approx \leq 5\%$) and there is no evidence of systematic or differential missingness between arms, a complete-case analysis may be reported as the primary analysis, with multiple imputation (MI) used as a sensitivity check. MI will be the primary approach if missingness is higher or shows evidence of association with treatment arm or baseline covariates [44–46]. Imputation models will include all outcome time-points, treatment allocation, baseline covariates, and auxiliary predictors. Appropriate conditional models will be applied by variable type (predictive mean matching for continuous, logistic/multinomial/ordinal regression for categorical). At least $m = 20$ imputations will be generated (increasing $m$ if the fraction of missing information is large). Chain convergence will be checked, and each imputed dataset will be analyzed with the pre-specified mixed model (or GEE for longitudinal outcomes). Estimates will be pooled using Rubin's rules. Robustness will be examined using complete-case analyses, pattern-mixture approaches, and delta-adjustment (tipping-point) sensitivity analyses. Implementation and diagnostics will be conducted in R.

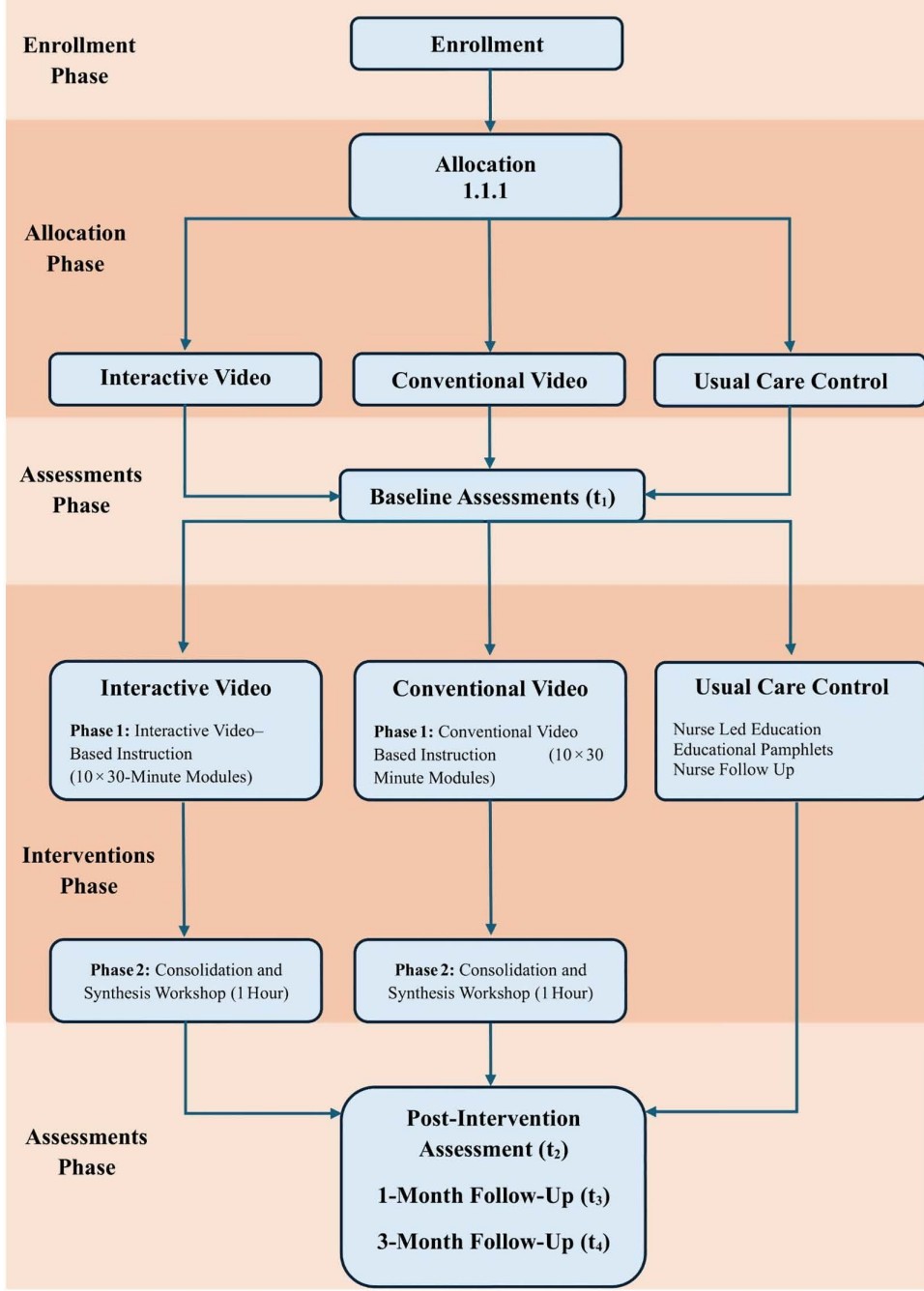

**Fig 3. Procedural flow of RCT protocol.**

All data will be entered into STATA version 17 and R software using a double-entry method to minimize errors. Coding will follow predefined guidelines, with range and consistency checks applied regularly. The research team will be the only people able to access the data, which will be stored securely on password-protected systems. Paper records will be kept in locked cabinets. Identifiable information will be separated from study data to ensure confidentiality. Data will be retained

for five years post-study, after which it will be securely destroyed. A detailed data management plan is available upon request.

## Statistical analysis

The baseline characteristics will be summarized using descriptive statistics (mean ± SD, frequency, and percentage). Primary and secondary outcomes will be analyzed using longitudinal data analysis methodologies, accommodating repeated measures collected at four time points (baseline, immediate post-intervention, one-month, and three-month follow-up) and across three intervention groups. Depending on the attributes and distribution of the dependent variables, consideration will be given to both mixed models for repeated measures (MMRM) and generalized estimating equations (GEE).

MMRM will be applied to model continuous normally distributed outcomes, incorporating fixed effects for group, time, and their interaction, with an appropriate covariance structure to account for within-subject correlations. GEE will be used as an alternative approach, particularly for outcomes with non-normal distributions or when robust population-averaged effects are of interest. MMRM and GEE will be chosen based on model fit and assumptions.

Analyses will be conducted under the intention-to-treat principle. Missing data will be addressed using multiple imputations, assuming they are randomly missing. Effect measures, CI, and a significance level of $p < 0.05$ will be reported. Adjusted analyses may include relevant covariates, with the primary analysis specified accordingly.

## Discussion

The present protocol outlines an RCT with three parallel groups to evaluate and compare the effectiveness of interactive versus conventional video-based education on patient activation, treatment adherence, and IDWG in HD patients. This study addresses significant gaps in existing knowledge and provides high-quality evidence to support educational interventions in this vulnerable population.

Patients undergoing HD encounter multifaceted difficulties in managing stage 5 CKD, necessitating rigorous compliance with stringent treatment protocols, encompassing pharmaceutical consumption, fluid and dietary limitations, and consistent dialysis appointments. Lack of compliance with these therapies could precipitate serious health issues, frequent inpatient care, and a marked decrease in one's well-being. Interventions of an educational nature play a pivotal role in fostering patient self-management and enhancing clinical outcomes [47].

Available systematic reviews have revealed that digital health interventions (DHIs) are associated with "small to large" gains in treatment adherence among dialysis patients. Nonetheless, the quality of extant evidence is often assessed as "low to moderate," and the quantity of included trials is limited, thereby emphasizing the necessity for more comprehensive investigation. In particular, gains in dietary adherence and fluid management have been described as only "moderate" and "small," respectively [48], suggesting that these domains may benefit from more specialized interventions and direct support. Moreover, a traditional educational study in CKD patients undergoing HD—despite reporting significant improvements in knowledge and adherence to fluid, dialysis, and medication regimens—found no statistically significant change in dietary adherence or pre-dialysis weight [47]. This indicates that knowledge enhancement alone is insufficient to alter complex behaviors such as dietary and fluid management, highlighting the need for more innovative approaches.

The present trial directly addresses these evidence gaps by comparing interactive video, conventional video, and routine education. We hypothesize that the inherently active nature of interactive videos—with personalized feedback, problem-solving scenarios, and opportunities for active response—will more effectively enhance patient activation and yield greater improvements in patient activation and treatment adherence (particularly in the challenging domains of dietary and fluid management) and, consequently, interdialytic weight control. Prior studies have demonstrated that videos incorporating advanced interactive features (e.g., domain-specific questions and tasks) are significantly more effective than those offering only basic navigation interactivity or none at all [49]. A recent meta-analysis reported an overall effect size of Hedges's g = 0.522 in favor of interactive video interventions, consistent with active learning theories that posit

superior outcomes for learner-engaged modalities [49]. Furthermore, interactive video learning—leveraging multimedia, engagement, and customized content—has been recognized as an effective instructional tool that enhances knowledge acquisition, motivation, and user engagement. This approach, therefore, offers a scalable, cost-effective strategy for patient education in clinical settings.

## Strengths and limitations

By employing a three-arm parallel RCT design, this study provides the highest level of evidence for determining the relative effectiveness of the educational interventions. Balancing observed and unobserved participant characteristics achieved through randomization minimizes bias and enables a precise evaluation of causal relationships between interventions and outcomes. This design allows for causal inference and minimizes bias [50]. Moreover, high-quality protocols promote proper trial conduct, minimize avoidable protocol amendments, and facilitate comprehensive evaluation of scientific and ethical considerations. Having a standard video group and routine care control enables precise assessment of the added value of interactive video beyond mere information delivery or usual care. This comparative analysis distinguishes authentic intervention outcomes from confounding variables, thus affording a more comprehensive understanding of the relative effectiveness of various educational methodologies. A rigorous literature review and content validation ensure the intervention is high-quality, thorough, and clinically relevant. Offering all training content to the control arm after outcome evaluations demonstrates our commitment to ethical research principles and ensures equitable access to beneficial interventions for all participants [51]. Evaluating outcomes immediately after the training sessions and at one- and three-month follow-ups allows for assessment of the intervention's durability. Long-term outcome evaluation is vital in clinical research to determine the persistence and reliability of treatments and inform clinical decision-making. This approach aids in understanding the longevity of behavioral and clinical changes and is essential for a comprehensive evaluation of intervention efficacy [52].

This study has several limitations, including the inability to fully blind participants to their assigned video intervention; however, data analysts will remain blinded to minimize bias. The generalizability of the findings from the selected hospitals affiliated with Shahid Beheshti University of Medical Sciences may be limited despite broad inclusion criteria. Additionally, while validated instruments will assess activation and adherence, they rely on self-report and may be influenced by recall or social-desirability bias. Another significant limitation is the potential for methodological contamination between study arms (e.g., participant-to-participant or staff-mediated sharing of intervention content), which may dilute actual intervention effects.

## Conclusion

This protocol explains a rigorous RCT intended to address crucial deficiencies in the present evidence concerning the utilization of interactive educational technologies among HD patients. This study compares interactive video, conventional video, and standard education to determine these interventions' effectiveness and evaluate their comparative performance. The study benefits from enhanced internal validity and greater potential for external application through incorporating validated outcome measures, several follow-up evaluations, and a stringent implementation strategy. Interactive video education, if validated, presents a scalable, patient-focused strategy for enhancing activation, adherence, and clinical results in HD.

## Supporting information

**S1 Table. The SPIRIT 2025 checklist of items to address in a randomized trial protocol.**
(DOCX)

**S2 File. Approved Protocol-EN.**
(DOCX)

**S3 File. Approved Protocol-Original.**
(DOC)

**S4 File. SPIRIT 2025 checklist.**
(DOCX)

## Author contributions

**Conceptualization:** Sogand Sarmadi, Neda Sanaie, Akbar Zare-Kaseb.

**Data curation:** Sogand Sarmadi, Akbar Zare-Kaseb.

**Formal analysis:** Sogand Sarmadi, Neda Sanaie, Akbar Zare-Kaseb.

**Funding acquisition:** Neda Sanaie, Akbar Zare-Kaseb.

**Investigation:** Sogand Sarmadi.

**Methodology:** Sogand Sarmadi, Neda Sanaie.

**Project administration:** Sogand Sarmadi, Neda Sanaie.

**Resources:** Akbar Zare-Kaseb.

**Software:** Sogand Sarmadi, Neda Sanaie.

**Supervision:** Akbar Zare-Kaseb.

**Validation:** Sogand Sarmadi, Neda Sanaie, Akbar Zare-Kaseb.

**Writing – original draft:** Sogand Sarmadi, Neda Sanaie, Akbar Zare-Kaseb.

**Writing – review & editing:** Sogand Sarmadi, Akbar Zare-Kaseb.

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
