## [Decision Letter · Decision Letter 0]

15 Sep 2025

PONE-D-25-42865
Comparing the Effects of Interactive and Conventional Video Education on Activation, Treatment Adherence and Weight Changes in Dialysis Patients: A Randomized Clinical Trial Protocol
PLOS ONE

Dear Dr. Zare-Kaseb,

Thank you for submitting your manuscript to PLOS ONE. After careful consideration, we feel that it has merit but does not fully meet PLOS ONE’s publication criteria as it currently stands. Therefore, we invite you to submit a revised version of the manuscript that addresses the points raised during the review process.
 
Both reviewers agree that this is a well-written protocol addressing an important question on improving patient activation and adherence in dialysis care through interactive and conventional video education. The study is relevant to the PLOS ONE scope and adheres to SPIRIT guidance. However, before the manuscript can be accepted, several issues must be addressed to ensure methodological rigour and clarity.

**Required Revisions:**

1.**Cluster randomisation with three hospitals: **The protocol assigns one hospital to each arm. With only three clusters, intervention effects may be confounded by site-level differences and cannot be reliably separated from cluster effects. Please provide a strong justification for this design, at least give detailed information on comparability of these 3 centers, acknowledge it explicitly as a limitation, and describe how intra-cluster correlation will be addressed in both the pilot-based sample size calculation and subsequent analysis.
 
2.**Sample size and pilot study:**
** **Reviewers requested further detail on the planned pilot. Please specify how the pilot will be conducted (sample, measures, anticipated effect size estimates) and clarify how these data will inform the main trial’s power calculation. In addition, explain the rationale for using 95% power, which is higher than common practice, and justify feasibility of achieving this with the proposed design.

**3**.**Instructor bias and workshop imbalance: **Reviewer #1 raised valid concerns that the Phase 2 workshops may introduce instructor-driven bias and are not provided to the control arm. Please describe standardisation measures (e.g., facilitator scripts, checklists) to minimise bias, and justify why the control arm does not receive an equivalent structured session.

**4.Blinding of outcome assessment** Reviewer #2 questioned why assessors are not blinded. Please provide justification for this choice and describe any measures you will use to minimise bias (e.g., independent assessors, scripted administration).

**5.**
**Content validation: **Clarify why ICU nurses and anaesthesia specialists were selected to review programme content rather than dialysis-experienced nurses or nephrologists. If possible, expand content validation to include specialists directly involved in haemodialysis care.

**6**.**Blinding of outcome assessment: **Reviewer #2 questioned why assessors are not blinded. Please provide justification for this choice and describe any measures you will use to minimise bias (e.g., independent assessors, scripted administration).

**7.Edits for clarity** Replace all placeholders (e.g., “XXXX” for ethics approval and registration codes) with the correct identifiers. Revise Figure 3 to ensure baseline assessment is clearly shown to occur before intervention delivery.
 
For the recommended revisions, the authors should provide more detail on missing data handling (including assumptions for multiple imputation, number of imputations, and planned sensitivity analyses), clarify that interdialytic weight gain reflects both adherence and clinical factors and specify how potential confounders such as residual renal function and comorbidities will be addressed, ensure consistency of terminology by defining acronyms such as “PAM” at first use, review and streamline the introduction to focus on the specific evidence gap rather than extensive background epidemiology, and undertake minor grammatical edits to improve concision and clarity. The authors should also address the reviewers’ comments point by point in their rebuttal to ensure all issues raised have been adequately resolved. 

We look forward to receiving your revised manuscript.

Kind regards,

Jeerath Phannajit, M.D, Ph.D.

Academic Editor

PLOS ONE

2. Please include a separate caption for each figure in your manuscript.

Reviewers' comments:

Reviewer's Responses to Questions

**Comments to the Author**

1. Does the manuscript provide a valid rationale for the proposed study, with clearly identified and justified research questions?

Reviewer #1: Yes

Reviewer #2: Yes

2. Is the protocol technically sound and planned in a manner that will lead to a meaningful outcome and allow testing the stated hypotheses?

Reviewer #1: Yes

Reviewer #2: Yes

3. Is the methodology feasible and described in sufficient detail to allow the work to be replicable?

Reviewer #1: Yes

Reviewer #2: Yes

4. Have the authors described where all data underlying the findings will be made available when the study is complete?

Reviewer #1: No

Reviewer #2: No

5. Is the manuscript presented in an intelligible fashion and written in standard English?

Reviewer #1: Yes

Reviewer #2: Yes

6. Review Comments to the Author

You may also provide optional suggestions and comments to authors that they might find helpful in planning their study.

Reviewer #1: This study protocol addresses an important and timely question on how best to enhance patient activation and adherence in hemodialysis through interactive versus conventional video education. The design is rigorous, with a three-arm, cluster-randomized trial framework, but there are several methodological issues that need clarification to ensure validity and interpretability.

Major issues

• How will you decrease potential instructor bias in the interactive video education arm in Phase 2, given that the instructor facilitates critical discussions, guides extraction of salient clinical concepts, and synthesizes group findings? This process may introduce intervention bias by directing participants to retain knowledge specific to the PAM questions.

• Why is the 1-hour workshop implemented only in the interactive video and conventional video arms, while the usual care control arm does not receive a workshop?

• How will you ensure content equivalence across interventions, particularly in terms of detail and knowledge structure?

• How will you address possible contamination between patients within the same dialysis facility or even the same dialysis session?

• In Figure 3, should baseline assessment occur before the intervention?

Minor issues

• The abbreviation “PAM tool” is not mentioned before page 4, paragraph 2.

• Please provide the ethics committee code (XXXX) and ClinicalTrials.gov registration number (XXXX).

• Why will the program content be assessed by ICU nurses rather than hemodialysis nurses?

• How were the ten faculty members who validated the program content selected?

• Why were anesthesia specialists involved as key evaluators of the program content, rather than nephrologists or internists?

Reviewer #2: The research team plans a clinical trial to assess the impact of interactive and conventional video-based education on patient activation, treatment adherence, and inter-dialytic weight gain in a dialysis population.

1. Sample size will be determined based on a pilot study. However, there is a lack of pilot study information.

2. Please clarify why to use 95% power for the sample size calculation as it is much higher than the commonly used level.

3. Please provide some more details on the missing data imputation

4. Please clarify why outcome accessors will not be blinded as this may introduce the bias.

5. Please provide the justification for the feasibility to implement the study within four months as planned.

7. PLOS authors have the option to publish the peer review history of their article (what does this mean?). If published, this will include your full peer review and any attached files.

Reviewer #1: No

Reviewer #2: No

---

## [Author Response · Author response to Decision Letter 1]

19 Sep 2025

Response to Required Revisions:

1.Cluster randomisation with three hospitals: The protocol assigns one hospital to each arm. With only three clusters, intervention effects may be confounded by site-level differences and cannot be reliably separated from cluster effects. Please provide a strong justification for this design, at least give detailed information on comparability of these 3 centers, acknowledge it explicitly as a limitation, and describe how intra-cluster correlation will be addressed in both the pilot-based sample size calculation and subsequent analysis.

Re: We appreciate this insightful comment. Following the reviewers’ and editor’s recommendations, we have revised the study design. Instead of assigning entire hospitals to a single intervention arm, we now use individual-level randomization stratified by hospital and dialysis shift. This ensures balance across sites, prevents confounding by hospital-level characteristics, and eliminates the limitations associated with only three clusters.

2.Sample size and pilot study: Reviewers requested further detail on the planned pilot. Please specify how the pilot will be conducted (sample, measures, anticipated effect size estimates) and clarify how these data will inform the main trial’s power calculation. In addition, explain the rationale for using 95% power, which is higher than common practice, and justify feasibility of achieving this with the proposed design.

Re: Thank you for this important request for clarification. We have expanded the manuscript to describe the pilot study and to correct our power statement.

3.Instructor bias and workshop imbalance: Reviewer #1 raised valid concerns that the Phase 2 workshops may introduce instructor-driven bias and are not provided to the control arm. Please describe standardisation measures (e.g., facilitator scripts, checklists) to minimise bias, and justify why the control arm does not receive an equivalent structured session.

Re: We agree with the reviewer and have addressed this by using facilitator scripts, fidelity checklists, and standardized training to minimize instructor bias. Selected sessions will also be recorded for independent review. The control group receives the standard nurse-led education routinely provided in these centers; an additional structured workshop was not included to preserve the contrast with usual care, but all educational content will be offered to this group after outcome assessment.

4.Blinding of outcome assessment Reviewer #2 questioned why assessors are not blinded. Please provide justification for this choice and describe any measures you will use to minimise bias (e.g., independent assessors, scripted administration).

Re: We thank the reviewer for this point. While participants and facilitators cannot be blinded due to the nature of the interventions, outcome assessors will remain blinded to group allocation. They will administer questionnaires using standardized scripts and will extract clinical data independently. In addition, all analyses will be conducted on de-identified datasets labeled only as Group A, B, or C until the primary analyses are complete. These measures are intended to minimize detection and analytic bias.

5.Content validation: Clarify why ICU nurses and anaesthesia specialists were selected to review programme content rather than dialysis-experienced nurses or nephrologists. If possible, expand content validation to include specialists directly involved in haemodialysis care.

 Re: Thank you for your instructive comment. This was a mistake and is now corrected.

6.Blinding of outcome assessment: Reviewer #2 questioned why assessors are not blinded. Please provide justification for this choice and describe any measures you will use to minimise bias (e.g., independent assessors, scripted administration).

 Re: We thank the reviewer for this point. While participants and facilitators cannot be blinded due to the nature of the interventions, outcome assessors will remain blinded to group allocation. They will administer questionnaires using standardized scripts and will extract clinical data independently. In addition, all analyses will be conducted on de-identified datasets labeled only as Group A, B, or C until the primary analyses are complete. These measures are intended to minimize detection and analytic bias

7.Edits for clarity Replace all placeholders (e.g., “XXXX” for ethics approval and registration codes) with the correct identifiers. Revise Figure 3 to ensure baseline assessment is clearly shown to occur before intervention delivery.

Re: Thanks for your valuable comment. This is now corrected.

For the recommended revisions, the authors should provide more detail on missing data handling (including assumptions for multiple imputation, number of imputations, and planned sensitivity analyses), clarify that interdialytic weight gain reflects both adherence and clinical factors and specify how potential confounders such as residual renal function and comorbidities will be addressed, ensure consistency of terminology by defining acronyms such as “PAM” at first use, review and streamline the introduction to focus on the specific evidence gap rather than extensive background epidemiology, and undertake minor grammatical edits to improve concision and clarity. The authors should also address the reviewers’ comments point by point in their rebuttal to ensure all issues raised have been adequately resolved.

We sincerely thank the editor and reviewer for these constructive suggestions, which have significantly improved the clarity and rigor of our manuscript. The following revisions have been made:

Missing data handling:

We have expanded the Statistical Analysis section to provide comprehensive details on missing data handling. Specifically, we now describe the assumptions for multiple imputation, the number of imputations, and the planned sensitivity analyses, including complete-case analysis, pattern-mixture models, and delta-adjustment approaches.

Interdialytic weight gain (IDWG):

We clarified that IDWG reflects not only fluid adherence but also clinical factors such as residual renal function and comorbidities.

Potential confounders:

We specified how residual renal function, comorbidities, and other relevant covariates will be recorded and incorporated into adjusted analyses to minimize potential confounding bias.

Consistency of terminology:

All acronyms are now defined at their first occurrence. For example, “Patient Activation Measure (PAM)” has been fully clarified at first use.

Introduction revision:

The Introduction has been streamlined to emphasize the evidence gap addressed by this trial and the novelty of our study, with less emphasis on broad epidemiological background.

Language and grammar:

Minor grammatical and stylistic edits have been undertaken throughout the manuscript to enhance concision, clarity, and readability.

---

## [Decision Letter · Decision Letter 1]

29 Sep 2025

Comparing the effects of interactive and conventional video education on activation, treatment adherence, and weight changes in dialysis patients: a randomized clinical trial protocol

PONE-D-25-42865R1

Dear Dr. Zare-Kaseb,

We’re pleased to inform you that your manuscript has been judged scientifically suitable for publication and will be formally accepted for publication once it meets all outstanding technical requirements.

Kind regards,

Jeerath Phannajit, M.D, Ph.D.

Academic Editor

PLOS ONE

Reviewers' comments:

Reviewer's Responses to Questions

**Comments to the Author**

1. Does the manuscript provide a valid rationale for the proposed study, with clearly identified and justified research questions?

Reviewer #1: Yes

Reviewer #2: Yes

2. Is the protocol technically sound and planned in a manner that will lead to a meaningful outcome and allow testing the stated hypotheses?

Reviewer #1: Yes

Reviewer #2: Yes

3. Is the methodology feasible and described in sufficient detail to allow the work to be replicable?

Reviewer #1: Yes

Reviewer #2: Yes

4. Have the authors described where all data underlying the findings will be made available when the study is complete?

Reviewer #1: Yes

Reviewer #2: No

5. Is the manuscript presented in an intelligible fashion and written in standard English?

Reviewer #1: Yes

Reviewer #2: Yes

6. Review Comments to the Author

You may also provide optional suggestions and comments to authors that they might find helpful in planning their study.

Reviewer #1: I have reviewed the revised protocol and I accept the revision. I appreciate the authors’ responsiveness.

Reviewer #2: Thank you for satisfactorily addressing all the raised comments. I have no further concerns on this manuscript.

7. PLOS authors have the option to publish the peer review history of their article (what does this mean?). If published, this will include your full peer review and any attached files.

Reviewer #1: No

Reviewer #2: No

---

## [Editor Report · Acceptance letter]

PONE-D-25-42865R1

PLOS ONE

Dear Dr. Zare-Kaseb,

I'm pleased to inform you that your manuscript has been deemed suitable for publication in PLOS ONE. Congratulations! Your manuscript is now being handed over to our production team.

Kind regards,

on behalf of

Dr. Jeerath Phannajit

Academic Editor

PLOS ONE